# Retinal Ischemic Perivascular Lesions Are Associated with Cardiovascular Diseases in Patients with Severe Carotid Artery Stenosis

**DOI:** 10.3390/jcm15010246

**Published:** 2025-12-29

**Authors:** Li Zhang, Ying-Ying Chen, Chun-Yan Lei, Fei-Peng Jiang, Tian-Yu Yang, Zhi-Hao Xiao, Sheng Gao, Mei-Xia Zhang

**Affiliations:** Department of Ophthalmology, West China Hospital, Sichuan University, Chengdu 610041, China; hx_zhang_li@163.com (L.Z.); chenyingying@scu.edu.cn (Y.-Y.C.); 13488934224@163.com (C.-Y.L.); feipengj@163.com (F.-P.J.); tianyuyang1104@163.com (T.-Y.Y.); xzh740611@163.com (Z.-H.X.); gaosheng322@163.com (S.G.)

**Keywords:** retinal ischemic perivascular lesions, cardiovascular diseases, carotid artery stenosis, swept-source optical coherence tomography

## Abstract

**Background/Objectives**: The objective of this study is to evaluate the association between retinal ischemic perivascular lesions (RIPLs) and cardiovascular diseases (CVDs) among patients diagnosed with severe carotid artery stenosis (SCAS). **Methods**: Consecutive patients (123 patients) with SCAS were included and underwent swept-source optical coherence tomography angiography (SS-OCTA). Retinal structural and blood flow parameters of the macular region were calculated and compared between patients with and without CVD. The prevalence of RIPLs confirmed on B-scan images was compared between patients with and without CVD. The relationship between RIPLs and CVD in patients with SCAS was explored using multivariate logistic regression analysis. Subgroup analysis was conducted to evaluate the relationship between ipsilateral RIPLs, contralateral RIPLs, and bilateral RIPLs and CVD. **Results**: Of the 123 patients with SCAS, 61 patients (49.6%) had a history of CVD. The CVD group had lower vessel density in the superficial and deep retinal vascular complexes, thinner inner retinal thickness, and thicker outer retinal thickness compared with the non-CVD group. A higher prevalence of RIPLs was found in the CVD group compared to the non-CVD group (55.7% vs. 30.6%, *p* = 0.006). The presence of RIPLs was significantly associated with CVD in SCAS patients (OR = 3.953 [1.695–9.219], *p* = 0.001) after adjusting for covariates. Subgroup analysis revealed greatest risk of bilateral RIPLs for CVD (OR = 7.383 [1.749–30.393], *p* = 0.006), followed by contralateral RIPLs (OR = 4.024 [1.432–11.306], *p* = 0.008) and ipsilateral RIPLs (OR = 2.951 [1.258–6.921], *p* = 0.013). **Conclusions**: The presence of RIPLs is significantly associated with CVD in SCAS patients. Findings of this research demonstrated that evaluation of RIPLs may help identify high-risk SCAS patients and facilitate special medical care for this group of patients.

## 1. Introduction

Cardiovascular diseases (CVDs) are a group of disorders of the heart and blood vessels which remain one of the greatest threats to the health of Chinese residents, accounting for 48.98% and 47.35% of deaths in rural and urban areas, respectively [1]. Globally, it has been estimated that 17.9 million people died from CVDs in 2019, and 23.6 million annual CVD deaths are predicted to occur by 2030 [2,3].

Retinal ischemic perivascular lesions (RIPLs) are focal atrophy of the inner nuclear layer (INL) identified on OCT images and were first termed in 2021 [4]. Acute ischemia of the retina initially develops at the level of the retinal deep capillary plexus, which manifests as paracentral acute middle maculopathy (PAMM). It is believed that RIPLs are the legacy of PAMM lesions [5]. Recently, RIPLs have been linked to various cardiovascular conditions, such as coronary artery disease [4], atrial fibrillation [6,7], myocardial infarction [8], cerebral small vessel disease burden [9], sickle cell maculopathy [10], carotid artery stenosis (CAS) [11,12,13], and even subclinical cardiovascular diseases [14]. CAS is a known risk factor for acute cardiovascular events and requires early intervention [15,16]. Recognizing high-risk populations among patients with CAS may facilitate better diagnosis and medical management but remains clinically challenging. The aim of this study was to determine if RIPLs are a marker of CVD in a cohort of patients with severe carotid artery stenosis (SCAS). It has been reported that retinal structural and blood flow traits measured by OCT and OCTA are non-invasive markers that reflect systemic vascular biology, so changes in retinal structure were also studied in this research.

## 2. Materials and Methods

Ethics statement and study design: This was a single-center, prospective, case–control study conducted at Sichuan University, West China Hospital, China. The study was designed and performed following the ethical tenets of the 1964 Helsinki Declaration and approved by the Institute Ethics Committee of West China Hospital with verifiable consent (Approval number: 20231171). Informed consent was obtained from all participants and/or their legal guardians. The details of the study design and patient recruitment have been reported elsewhere [17]. Briefly, participants with carotid artery stenosis ≥ 70% were included in this study. The participant exclusion criteria were as follows: (1) presented retinopathy other than RIPLs, such as epiretinal membranes, macular holes, retinal artery/vein occlusion, diabetic retinopathy, amblyopia, macular edema, uveitis, etc.; (2) previously received any fundus treatment such as retinal laser, intravitreal injection, vitrectomy, etc.; (3) axial length > 26 mm or mean spherical equivalent < −6 diopters; (4) complicated with neurodegenerative or demyelinating diseases, such as Alzheimer’s disease, Parkinson’s disease, or multiple sclerosis; and (6) poor quality of OCTA images (image quality ≤ 6).

Data collection: Individual medical records were reviewed, and information on age, gender, BMI, blood pressure, smoking and drinking history, prior diagnosis of hypertension, diabetes, and dyslipidemia was recorded. Subjects were classified into the CVD group if they had a diagnosis of coronary heart disease, congestive heart failure, myocardial infarction, atrial fibrillation, or stroke based on evaluations by primary care physicians, and others were classified into the control group.

Image analysis: All patients underwent swept-source OCTA (SS-OCTA) imaging covering an area of 16 × 16 volume using a 400 kHz SS-OCTA instrument (TowardPi BMizar; TowardPi Medical Technology, Beijing, China). With an OCTA scan, 1024 OCT B-scans were generated. For each patient, RIPLs were assessed with 1024 OCT images within the range of 16 × 16 volume covered by the OCTA scan. RIPLs were defined as the presence of focal atrophy or thinning of the INL with expanded hyporeflectivity of the ONL on OCT B-scan images. All images were reviewed by 2 independent and masked trained graders. Any discrepancies were resolved by discussion and determined by a third grader.

Retinal structure and blood flow assessment: Topographic measurements were performed using the built-in software of Early Treatment Diabetic Retinopathy Study (ETDRS) subfield analysis (Figure 1). The acquired data were automatically averaged through the following subfields and sectors: the central fovea subfield within the inner 1 mm diameter circle; inner ring: the circle subfield between the 1–3 mm diameter circle and 3–6 mm diameter circle; and outer ring: the circle subfield between the 3–6 mm diameter circle and 6–9 mm diameter circle. The inner and outer circles were sectioned into nasal, temporal, superior, and inferior quadrants. The macular region was finally divided into 9 subfields including the central macula (CM), superior of the inner/outer ring (SIR/SOR), inferior of the inner/outer ring (IIR/IOR), nasal of the inner/outer ring (NIR/NOR), and temporal of the inner/outer ring (TIR/TOR). Retinal structural parameters including the inner retinal thickness (from inner limiting membrane to inner nuclear layer) and outer retinal thickness (from outer plexiform layer to retinal pigment epithelium layer) and retinal blood flow parameters including the vessel density of the retinal superficial vascular complex (VDRSVC) and vessel density of the retinal deep vascular complex (VDRDVC) were calculated automatically by the device.

Statistical analysis: All data were processed using SPSS version 26 (SPSS, Inc., Chicago, IL, USA). Mean and standard deviation were used for describing continuous variables. Frequency with percentages was used for describing categorical variables. Statistical comparisons between groups were performed using independent-sample *t*-tests and Chi-square tests. The concordance between graders was assessed using the kappa coefficient. A multivariate logistic regression model with stepwise backward selection was used to evaluate the relationship between RIPLs and CVD. Covariates included age, gender, BMI, smoking/drinking status, hypertension, diabetes, and dyslipidemia. *p* values < 0.05 were considered statistically significant.

## 3. Results

**Demographic characteristics:** OCTA images of 155 patients with severe carotid artery stenosis were acquired between September 2023 and January 2024. After excluding patients with prior retinopathy other than RIPLs, a total of 246 eyes from 123 patients (109 males and 14 females) with SCAS were included in the imaging analysis with a mean age of 66.01 ± 24.0 years and mean BMI of 24.0 ± 2.68. Among them, 61 patients (49.6%) presented with CVD, 50 (40.7%) with diabetes mellitus, 82 (66.7%) with hypertension, and 23 (18.7%) with hyperlipidemia. Seventy-two patients (58.5%) had a history of smoking, and fifty-nine patients (48.0%) had a history of drinking (Table 1).

We identified 61 patients with CVD as cases and 62 patients without CVD as controls. The baseline characteristics for cases and controls are shown in Table 2. There was no difference between the two groups in terms of age, sex distribution, DM, HT, HL, smoking, drinking, visual acuity, or visual symptoms. The BMI was significantly greater in patients with CVD (24.60 ± 2.79) than in patients without CVD (23.41 ± 2.56, *p* = 0.013).

**Retinal structural and blood flow assessment:** As shown in Table 3, compared with patients without CVD, the inner retinal thickness of patients with CVD in the subfields of NIR (122.6 mm ± 14.5 vs. 128.0 mm ± 13.6, *p* = 0.035) and TOR (100.3 mm ± 10.7 vs. 105.4 mm ± 14.0, *p* = 0.024) was significantly lower. The outer retinal thickness in the subfield of IOR was significantly higher in the CVD group compared with the non-CVD group (198.7 mm ± 24.3 vs. 191.0 mm ± 15.2, *p* = 0.037). The VDRSVC in the subfields of IIR (37.4 ± 3.9 vs. 40.5 ± 2.9, *p* < 0.001), TOR (38.1 ± 3.0 vs. 40.9 ± 3.1, *p* < 0.001), and IOR (41.5 ± 3.4 vs. 43.8 ± 3.7, *p* = 0.001) was significantly lower in the CVD group compared with the non-CVD group. The VDRDVC in the subfield of IIR (39.0 ± 4.4 vs. 42.4 ± 4.4, *p* < 0.001) was significantly lower in the CVD group compared with the non-CVD group.

**Identification of retinal ischemic perivascular lesions (RIPLs):** After reviewing all the B-scan and blood flow images, we identified a total of 53 patients (43.1%) presenting with RIPLs (Figure 2). Of them, 39 patients (31.7%) had ipsilateral RIPLs, 29 patients (23.6%) had contralateral RIPLs, and 16 patients (13.0%) had bilateral RIPLs (Table 4). The inter-rater agreement between the two original graders for the detection of RIPLs was 85.4% with a kappa coefficient of 0.70 (*p* < 0.001).

**Association between RIPLs and CVD:** The percentage of subjects with RIPLs, ipsilateral RIPLs, contralateral RIPLs, and bilateral RIPLs was higher in patients with CVD compared with patients without CVD (55.7% versus 30.6%, *p* = 0.006; 42.6% vs. 30.0%, *p* = 0.012, 31.1% vs. 16.1%, *p* = 0.05, 21.3% vs. 4.8%, *p* = 0.007).

In a multivariable logistic regression model adjusted for covariates, age, BMI, and the presence of RIPLs were significantly associated with CVD (OR = 1.053 [1.003–1.106], *p* = 0.039; OR = 1.259 [1.063–1.490], *p* = 0.008; OR = 3.953 [1.695–9.219], *p* = 0.001). Subgroup analysis revealed that bilateral RIPLs, contralateral RIPLs, and ipsilateral RIPLs were significantly associated with CVD (OR = 7.383 [1.749–30.393], *p* = 0.006; OR = 4.024 [1.432–11.306], *p* = 0.008; OR = 2.951 [1.258–6.921], *p* = 0.013), with bilateral RIPLs showing the greatest risk (Table 5, Figure 3).

## 4. Discussion

In the present study, we found lower inner retinal thickness, higher outer retinal thickness, and lower VDRSVC and VDRDVC in SCAS patients with CVD compared with patients without CVD. We also demonstrated a higher prevalence of RIPLs, as detected by OCT imaging, in patients with CVD compared to those without in a cohort of individuals with SCAS. Specifically, in the subgroup analysis, bilateral RIPLs showed the greatest risk, followed by contralateral RIPLs and ipsilateral RIPLs.

The association between RIPLs and CVD has been evaluated in previous studies. Christopher et al. reported a higher number of RIPLs in patients with documented cardiovascular disease compared to healthy controls, and each RIPL was associated with an odds ratio of having cardiovascular disease of 1.60, and individuals with higher 10-year atherosclerotic cardiovascular disease scores had more RIPLs [4]. Another study by them reported newly diagnosed cardiovascular disease in 72.7% of cases with priorly documented RIPLs [14]. Recently, a prospective study also found a higher prevalence of RIPLs in patients with ischemic CVD or risk factors compared to healthy controls [18]. The association between RIPLs and CAS has also been evaluated in several studies. David et al. reported a higher mean number of RIPLs in patients with CAS compared to age-matched controls [11,13]. In their study, a high incidence of RIPLs of 91% was reported in a cohort of 22 CAS patients (11 had medically managed <60% stenosis, and 11 had a history of CAS treated with carotid endarterectomy). Another study found an association between RIPL presence, number, and distribution and the stenotic degree in patients with CAS, and an incidence of RIPL presence of 52% was reported in patients with 90–99% stenosis [12].

In the present study, we recruited CAS patients with stenosis > 70% which were at high risk of developing cardiovascular events and investigated the association between RIPLs and CVD in this specific group of patients, which was important for identifying high-risk patients and guiding further medical management. We found an overall RIPL incidence of 42.3% in patients with stenosis > 70%. The results of multivariable logistic regression analysis revealed that the presence of ipsilateral, contralateral, and bilateral RIPLs was associated with an increased risk of CVD, indicating the importance of screening for cardiovascular conditions in the management of high-risk CAS patients with RIPLs. Bilateral RIPLs showed the greatest risk, followed by contralateral and ipsilateral RIPLs. This emphasizes the necessity of evaluating both eyes in risk assessment. To relieve the ipsilateral compromised cerebral vascular condition, the blood flow of the contralateral hemisphere would be directed to the ipsilateral hemisphere via the anterior communicating artery, which means unilateral CAS may cause bilateral ocular ischemia. In a recent study [12], researchers also found increased contralateral RIPLs with an increased stenotic degree in patients with CAS, showing the effect of decreased ipsilateral ocular perfusion on the fellow eye. Age and BMI were also found to be associated with CVD in our study, which were reported risk factors for CVD in previous studies [19]. However, recently, a study by Bellanda et al. [20], which prospectively included a total of 559 patients, demonstrated no significant association between RIPLs and a history of major cardiac adverse events. Their sample consisted of patients presenting to a cardiology clinic, and they found an overall RIPL prevalence of 26.3%, which differed from our sample of patients with SCAS, which was a highly selected group of patients with an overall RIPL prevalence of 43.1%. This may indicate that the association between RIPLs and cardiovascular risk does not extend to a broader population of patients.

Ocular microvasculature is thought to have identical physiological and anatomical features to cerebral and coronary microcirculation. It has been shown that retinal vascular alterations could reflect systemic processes including cardiovascular diseases [21]. Retinal vascular features such as retinal nerve fiber thickness, retinal vessel density, the foveal avascular zone, and the retinal vascular fractal dimension have been linked to cardiovascular risks [22,23,24,25]. In the present study, we found decreased VDRSVC and VDRDVC in CVD patients, which is consistent with previous studies. However, we also detected thinner inner retinal thickness and thicker outer retinal thickness in patients with CVD, which was a new finding. RIPLs are considered to be the legacy of PAMM lesions and refer to a characteristic focal thinning of the INL because of retinal ischemia which initially affects the deep retinal vascular complex. However, unlike outer retinal atrophy in retinal artery occlusion, a compensatory upward expansion of the ONL is presented in RIPLs, possibly to increase the ability to receive light signals. This may be responsible for the increased outer retinal thickness found in the present study. These changes were consistent with our findings of retinal changes in patients with CVD, who have a higher presence of RIPLs.

We have to acknowledge the limitations of this study. Firstly, this is a cross-sectional study, which was unable to assess the predictive value of RIPLs for the development of cardiovascular events, and this should be further investigated in future studies. Secondly, the diagnoses of CVDs were based on chart reviews, which may be inaccurate and lack consistency. Thirdly, the number of RIPLs in each patient was not assessed, which should be explored in future studies.

## 5. Conclusions

This study investigated the association between RIPLs and CVD in SCAS patients, in a highly selected high-risk cohort, and established a link between RIPLs and CVD in patients with SCAS. These findings emphasize the utility of RIPL detection in the risk assessment of cardiovascular conditions for patients with SCAS.

## Figures and Tables

**Figure 1 jcm-15-00246-f001:**
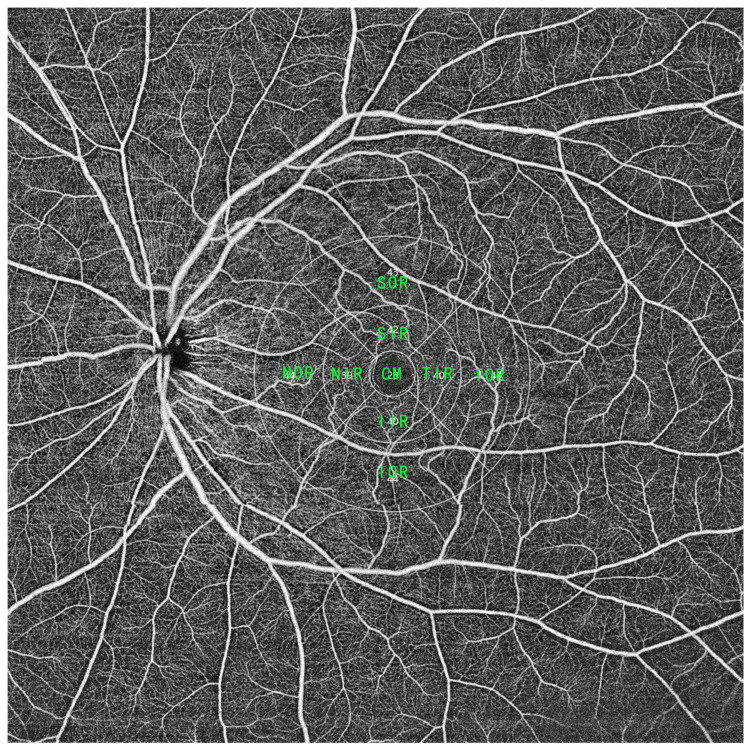
A widefield SS-OCTA scan within a range of 16 × 16 mm. To analyze macular blood flow change, the macular region (central macular with a bandwidth of 1 mm and two surrounding rings with a bandwidth of 3 mm) was divided into 9 parts, including the central macular (CM), superior of the inner/outer ring (SIR/SOR), inferior of the inner/outer ring (IIR/IOR), nasal of the inner/outer ring (NIR/NOR), and temporal of the inner/outer ring (TIR/TOR).

**Figure 2 jcm-15-00246-f002:**
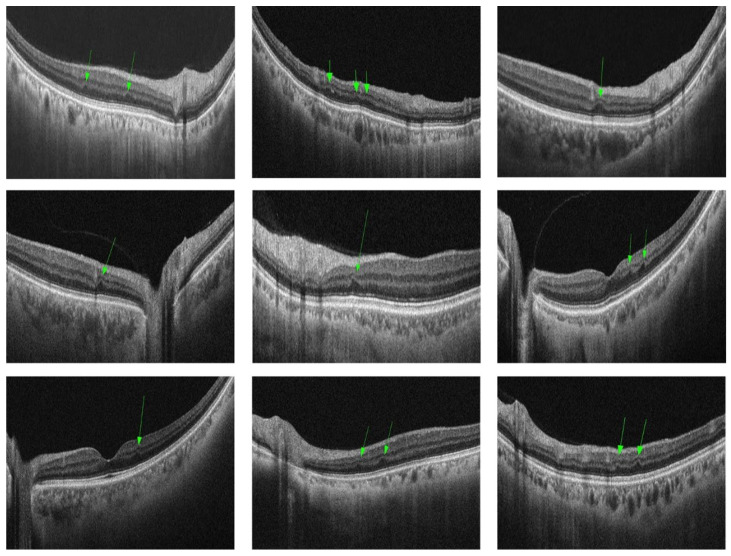
Retinal ischemic perivascular lesions (RIPLs) detected in different patients with severe carotid artery stenosis (SCAS) on swept-source optical coherence tomography (SS-OCT) B-scans, characterized by focal thinning of the inner nuclear layer and expansion of the outer nuclear layer (green arrows).

**Figure 3 jcm-15-00246-f003:**
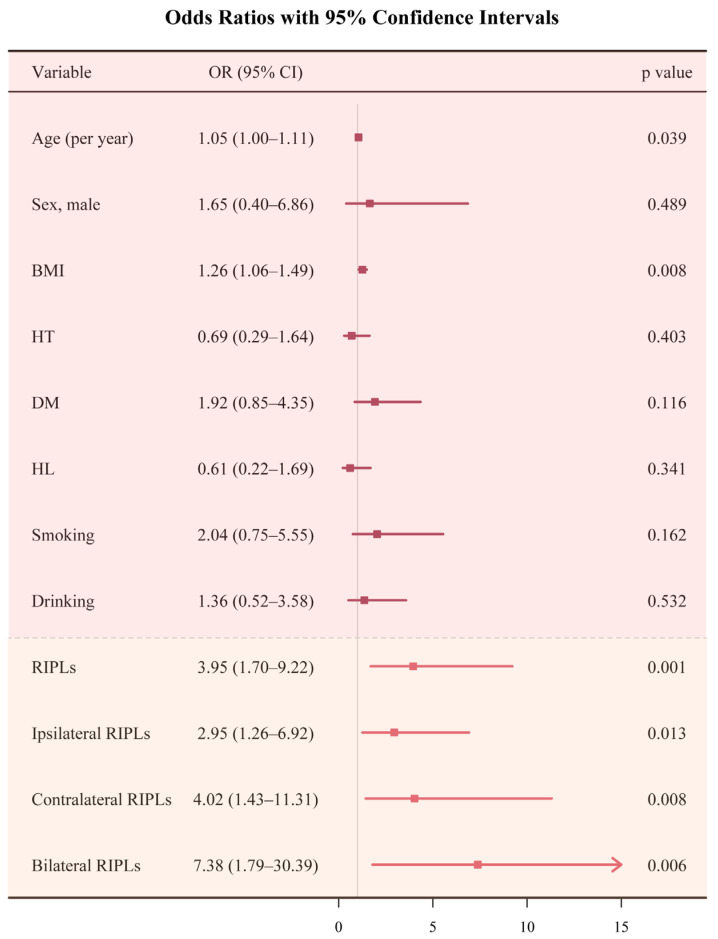
Forest plot of associations between variables and CVD in patients with severe carotid artery stenosis (SCAS).

**Table 1 jcm-15-00246-t001:** Demographic characteristics of 123 included patients with SCAS.

	Patients
No.	123
Age (years)	66.01 ± 24.0
Sex	109M (88.6%), 14F (11.4%)
CVD	61 (49.6%)
DM	50 (40.7%)
HT	82 (66.7%)
HL	23 (18.7%)
Smoking	72 (58.5%)
Drinking	59 (48.0%)
BMI	24.0 ± 2.68

SCAS: severe carotid artery stenosis; CVD: cardiovascular disease; DM: diabetes mellitus; HT: hypertension; HL: hyperlipidemia; BMI: body mass index.

**Table 2 jcm-15-00246-t002:** Baseline characteristics of SCAS patients with or without CVD.

	CVD Group	Non-CVD Group	*p* Value
No.	61	62	
Age (years)	67.03 ± 8.24	65.00 ± 9.67	0.378
Male	54 (88.5%)	55 (88.7%)	0.974
DM	30 (49.2%)	20 (32.3%)	0.056
HT	41 (67.2%)	41 (66.1%)	0.899
HL	11 (18.0%)	12 (19.4%)	0.851
Smoking	39 (63.9%)	33 (53.2%)	0.228
Drinking	31 (50.8%)	28 (45.2%)	0.530
BMI	24.56 ± 2.71	23.47 ± 2.56	0.503
RIPLs	34 (55.7%)	19 (30.6%)	0.006 *
Ipsilateral RIPLs	26 (42.6%)	13 (30.0%)	0.012 *
Contralateral RIPLs	19 (31.1%)	10 (16.1%)	0.050 *
Bilateral RIPLs	13 (21.3%)	3 (4.8%)	0.007 *

SCAS: severe carotid artery stenosis; CVD: cardiovascular disease; DM: diabetes mellitus; HT: Hypertension; HL: hyperlipidemia; BMI: body mass index; RIPLs: retinal ischemic perivascular lesions; *: *p* < 0.05.

**Table 3 jcm-15-00246-t003:** Comparison of retinal structure and blood flow parameters between CVD and non-CVD groups.

CVD Group		Non-CVD Group	*p* Value	CVD Group	Non-CVD Group	*p* Value
**Inner Retinal Thickness**				**Outer Retinal Thickness**		
CM	59.4 ± 12.4	61.7 ± 12.0	0.315	236.4 ± 18.7	231.9 ± 25.7	0.272
TIR	118.2 ± 11.7	122.0 ± 12.9	0.084	232.1 ± 14.4	228.7 ± 19.3	0.268
SIR	128.6 ± 19.6	133.5 ± 19.9	0.169	231.9 ± 16.0	227.3 ± 13.9	0.091
NIR	122.6 ± 14.5	128.0 ± 13.6	0.035 *	240.4 ± 19.8	235.2 ± 24.0	0.197
IIR	130.8 ± 27.9	128.1 ± 16.0	0.510	230.7 ± 22.4	225.7 ± 19.2	0.182
TOR	100.3 ± 10.7	105.4 ± 14.0	0.024 *	209.5 ± 11.5	207.9 ± 12.1	0.467
SOR	115.5 ± 20.8	118.1 ± 15.1	0.431	207.7 ± 15.2	203.9 ± 11.4	0.118
NOR	127.7 ± 14.6	125.2 ± 15.9	0.360	208.9 ± 16.4	210.4 ± 17.5	0.621
IOR	111.7 ± 21.4	114.1 ± 20.2	0.503	198.7 ± 24.3	191.0 ± 15.2	0.037 *
**VDRSVC**				**VDRDVC**		
CM	32.1 ± 4.6	32.1 ± 4.6	0.955	30.8 ± 12.0	29.4 ± 10.6	0.494
TIR	39.4 ± 4.0	39.8 ± 3.8	0.591	41.4 ± 3.9	40.8 ± 5.4	0.529
SIR	40.3 ± 5.7	40.5 ± 4.3	0.781	41.5 ± 5.0	41.2 ± 4.9	0.740
NIR	38.9 ± 4.4	39.4 ± 5.7	0.597	40.8 ± 4.8	40.4 ± 7.7	0.742
IIR	37.4 ± 3.9	40.5 ± 2.9	<0.001 *	39.0 ± 4.4	42.4 ± 4.4	<0.001 *
TOR	38.1 ± 3.0	40.9 ± 3.1	<0.001 *	40.8 ± 4.5	40.5 ± 6.0	0.726
SOR	42.5 ± 4.5	42.7 ± 3.6	0.853	41.1 ± 5.1	40.2 ± 6.1	0.367
NOR	42.3 ± 3.3	42.1 ± 5.0	0.749	41.6 ± 3.6	41.1 ± 7.2	0.612
IOR	41.5 ± 3.4	43.8 ± 3.7	0.001 *	41.1 ± 4.5	40.8 ± 6.2	0.778

CVD: cardiovascular disease; VDRSVC: vessel density of the retinal superficial vascular complex; VDRDVC: vessel density of the retinal deep vascular complex; CM: central macular; TIR: temporal of the inner ring; SIR: superior of the inner ring; NIR: nasal of the inner ring; IIR: inferior of the inner ring; TOR: temporal of the outer ring; SOR: superior of the outer ring; NOR: nasal of the outer ring; IOR: inferior of the outer ring; *: *p* < 0.05.

**Table 4 jcm-15-00246-t004:** RIPLs in patients with SCAS.

	Number (Percentage)
Patients with RIPLs	53 (43.1%)
Ipsilateral RIPLs	39 (31.7%)
Contralateral RIPLs	29 (23.6%)
Bilateral RIPLs	16 (13.0%)

RIPLs: retinal ischemic perivascular lesions; SCAS: severe carotid artery stenosis.

**Table 5 jcm-15-00246-t005:** Multivariable logistic regression model evaluating the relationship between presence of RIPLs and CVD.

Variables	OR	95%CI	*p* Value
Age (per year)	1.053	1.003–1.106	0.039 *
Sex, male	1.652	0.398–6.859	0.489
BMI	1.259	1.063–1.490	0.008 *
HT	0.692	0.292–1.640	0.403
DM	1.923	0.851–4.346	0.116
HL	0.608	0.219–1.691	0.341
Smoking	2.040	0.750–5.547	0.162
Drinking	1.361	0.518–3.575	0.532
RIPLs	3.953	1.695–9.219	0.001 *
Ipsilateral RIPLs	2.951	1.258–6.921	0.013 *
Contralateral RIPLs	4.024	1.432–11.306	0.008 *
Bilateral RIPLs	7.383	1.794–30.393	0.006 *

RIPLs: retinal ischemic perivascular lesions; CVD: cardiovascular disease; OR: odds ratio; CI: confidence interval; BMI: body mass index; HT: hypertension; DM: diabetes mellitus; HL: hyperlipidemia; *: *p* < 0.05.

## Data Availability

The datasets used and/or analyzed during the current study are available from the corresponding author on reasonable request.

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
