# Peer review of "Retinal Ischemic Perivascular Lesions Are Associated with Cardiovascular Diseases in Patients with Severe Carotid Artery Stenosis"

_jcm, 2025, doi:10.3390/jcm15010246_

Round 1

Reviewer 1 Report

Comments and Suggestions for Authors

The present manuscript is very interesting from an imaging perspective because it introduces the concept of retinal ischemic perivascular lesions and discusses how they can be detected non-invasively. Even more importantly, it highlights their potential role in the early identification of patients at risk for cardiovascular disease.  I have several comments to make that could improve the clarity and quality of the manuscript.

  1. Please translate the supplementary file in English.
  2. What was the acquisition protocol for the OCT scans?
  3. Over how many OCT scans were the RIPLs assessed per patient?
  4. How many RIPLs were identified per patient?
  5. Did you attempt to determine whether there were focal changes in vascular density exactly at the location of each RIPL?
  6. RIPL are stronger associated with SCAS or CVD?
  7. How do you explain de presence of contralateral RIPLs in non-CVD?
  8. How do you physiopathologically explain the increased outer retinal thickness?

Author Response

Response to Reviewer 1 Comments

Dear reviewer 1:

Thank you for your precious comments and advice. Those comments are all valuable and very helpful for revising and improving our paper, as well as the important guiding significance to our research. We have read your comments carefully and made corresponding corrections which we hope could meet with approval. Please find the detailed responses below and the corresponding revisions are highlighted in the re-submitted files:

reviewer's suggestion, question, or comment

Author response

Line numbers

1.      Please translate the supplementary file in English.

Thank you for pointing this out. The supplementary file has been translated into English and resubmitted.

/

2.      What was the acquisition protocol for the OCT scans?

OCT images were obtained by OCTA scans. With an OCTA scan covering an area of 16*16 volume, 1024 OCT b-scans were generated. For each patient, RIPLs were assessed with 1024 OCT images within the range of 16*16 volume covered by the OCTA scan.

Line

76-78

3.      Over how many OCT scans were the RIPLs assessed per patient?

4.      How many RIPLs were identified per patient?

A previous study has found that the number of RIPLs in individuals with intermediate and high 10-year ASCVD (atherosclerotic cardiovascular disease risk) scores was higher than in those with low ASCVD risk scores [1]. However, we admit that the number of RIPLs in each patient was not assessed, which was a limitation of the present study. We have emphasized this limitation in the revised manuscript and this may be explored in future studies.

[1]. Long, C. P., Chan, A. X., Bakhoum, C. Y., Toomey, C. B., Madala, S., Garg, A. K., Freeman, W. R., Goldbaum, M. H., DeMaria, A. N., & Bakhoum, M. F. (2021). Prevalence of subclinical retinal ischemia in patients with cardiovascular disease - a hypothesis driven study. EClinicalMedicine33, 100775. https://doi.org/10.1016/j.eclinm.2021.100775

Line

251-252

5.      Did you attempt to determine whether there were focal changes in vascular density exactly at the location of each RIPL?

Actually, as is shown in the figure below, we have found dark areas in both blood flow and enface OCTA images exactly at the location of RIPLs, which was the result of decreased vessel density of deep vascular complex and atrophy of the inner nuclear layer (INL).

/

6.      RIPL are stronger associated with SCAS or CVD?

This is a very interesting but unanswered question. In the previous studies, researchers have found that patients with CAS or CVD had more RIPLs compared with controls. However, covariates were not similarly adjusted in these studies, so the odds ratios were incomparable. On one hand, CAS is a risk factor for developing acute cardiovascular events. On the other hand, CAS and CVD share the pathological basis of arteriosclerosis. Besides, CAS and CVD could lead to decreased ocular perfusion and the development of ocular ischemic lesions, which may explain increased RIPLs in these patients. In the present study, we recruited SCAS patients with and without CVD to explore ocular biomarkers of high-risk populations and found that CVD increased the presence of RIPLs in the background of SCAS, which showed additive effects. However, it’s difficult for us to draw a conclusion on the scientifical question you brought.

/

7.      How do you explain de presence of contralateral RIPLs in non-CVD?

To relieve the ipsilateral compromised cerebral vascular condition, blood flow of the contralateral hemisphere would be directed to the ipsilateral hemisphere via the anterior communicating artery, which may result in contralateral ocular ischemia. In a recent study, the researchers also found increased contralateral RIPLs with increased stenotic degree in patients with CAS [2], showing the effect of decreased ipsilateral ocular perfusion on the fellow eye.

[2]. Wang, H., Cao, L., Kwapong, W. R., Liu, R., Yan, Y., Wan, J., Liu, G., Hu, F., & Wu, B. (2025). RETINAL ISCHEMIC PERIVASCULAR LESIONS ARE ASSOCIATED WITH INCREASED CAROTID ARTERY STENOTIC DEGREE. Retina (Philadelphia, Pa.), 45(4), 748–755. https://doi.org/10.1097/IAE.0000000000004354 

Line

214-220

8.      How do you physiopathologically explain the increased outer retinal thickness?

RIPLs are considered the legacy of PAMM lesions and refer to a characteristic focal thinning of the INL because of retinal ischemia which initially affects the deep retinal vascular complex. However, unlike outer retinal atrophy in retinal artery occlusion, a compensatory upward expansion of the ONL is presented in RIPLs, possibly to increase the ability of receiving light signals. This may be responsible for the increased outer retinal thickness found in the present study.

Line

237-242

Reviewer 2 Report

Comments and Suggestions for Authors

The authors evaluated RIPLs as a possible predictive sign for CVD. Recently several studies have investigated the relevance of RIPLs and came to controversial conclusions. 

I would suggest, that the authors also incorporate critical studies (e.g. Bellanda et al, 10.1016/j.oret.2025.09.002) in the discussion and be even more cautious with the interpretation of the relevance of the results.

Author Response

Dear reviewer 2:

Thank you for your precious comments and advice. Those comments are all valuable and very helpful for revising and improving our paper, as well as the important guiding significance to our research. We have read your comments carefully and made corresponding corrections which we hope could meet with approval. Please find the detailed responses below and the corresponding revisions are highlighted in the re-submitted files:

reviewer's suggestion, question, or comment

Author response

Line numbers

1.      The authors evaluated RIPLs as a possible predictive sign for CVD. Recently several studies have investigated the relevance of RIPLs and came to controversial conclusions. 

I would suggest, that the authors also incorporate critical studies (e.g. Bellanda et al, 10.1016/j.oret.2025.09.002) in the discussion and be even more cautious with the interpretation of the relevance of the results.

We thank the reviewer for the valuable suggestion. We have carefully reviewed the recent publication on RIPLs and discussed their findings in the revised manuscript. Their sample consists of patients presenting to a cardiology clinic and found an overall RIPLs prevalence of 26.3%, which differed from our sample of patients with SCAS, which was a highly selected group of patients and with an overall RIPLs prevalence of 43.1%. This may indicate the association of RIPLs and cardiovascular risk does not extend to a broader population of patients.

Line

221-228
